# Study on the Effects and Mechanism of Temperature and AO Flux Density in the AO Interaction with Upilex-S Using the ReaxFF Method

Shiying Qiao [1], Haifu Jiang [1], Ruiqiong Zhai [1], Yuming Liu [1], Tao Li [2], Yanlin Xu [1] and Lixiang Jiang [1,*]

[1] Beijing Institute of Spacecraft Environment Engineering, Beijing 100094, China; alicyor@163.com (S.Q.); haifujiang@163.com (H.J.); lingid_83@163.com (R.Z.); lyming2005@126.com (Y.L.); xuyanlinbeijing@163.com (Y.X.)

[2] China Aerospace Science and Technology Corporation, Beijing 100048, China; litaohit@163.com

[*] Correspondence: jlx8972@163.com

**Abstract:** Atomic oxygen (AO), which is one of the most predominant and conspicuous space environmental factors in the low earth orbit, leads to severe deterioration of polymeric materials in spacecraft. The AO flux density and ambient temperature vary while a craft orbits in space; thus, it is necessary to pay close attention to the flux density and temperature effects on the mechanism of the AO interaction with materials. In past years, polyimide has been widely used on spacecraft due to its excellent performance—that is the reason why we chose Upilex-S as the object for study. It was analyzed using the ReaxFF reactive force field molecular dynamics simulation, respectively from the aspect of impact-induced temperature variation, mass loss, reaction product and erosion yield. The results show that dense AO deposition on the surface impedes further erosion at low temperatures, and the AO interaction with Upilex-S is exacerbated as the ambient temperature increases. However, the accelerating rate is inversely proportional to the ambient temperature, which means the higher the ambient temperature is, the slower it increases. On the other hand, the interaction rate of AO induced to Upilex-S is aggravated as the flux density increases at the lower stage, while the interaction rate begins to drop as the flux density increases at the higher level. The AO erosion effect is a complicated process rather than a simple summation of single atomic oxygen interactions. Our study could be used as a technical reference for the wide usage of Upilex-S on spacecraft.

**Keywords:** low earth orbit (LEO); atomic oxygen (AO); Upilex-S; lammps; ReaxFF

## 1. Introduction

Atomic oxygen (AO), which largely exists in the low earth orbit (LEO) environment, always grabs a researcher's attention as it induces serious degradation of space material. Plenty of scientific reports tell us that, although the kinetic energy of AO itself is not that noteworthy, AO bombardment energy can reach 4.5–5 eV with a fierce oxidizing property while spacecrafts orbit at 7–8 km/s in space [1]. Because of this, materials are vulnerable in space, especially polymers [2–5]. Among these, polyimide is popular and widely used around a spacecraft's surface, structure and thermal control module due to its excellent performance in mechanical properties, insulation properties, thermal stability, ultraviolet stability, light-weight advantages, etc.; however, it can be damaged so easily in the AO environment [6,7]. Over the years, quantities of AO environment tests have been conducted to aggravate polyimide's degradation so as to extend the spacecraft lifespan, either using in situ flying tests [8–10] or in ground-based test facilities [11–13]. Nevertheless, the dynamic procedure of the AO interaction and the atomical bonding and breaking could never be observed using those two methods. Furthermore, it is more challenging and costly to improve the test facility either in space or on the ground to promote authenticity and accuracy.

On the other hand, along with the progression in computing performance and related theoretical research, molecular dynamics simulation is becoming an additional and alternative option that can be used to conduct research at the atomic micro-scaled level. In the research of molecular dynamics simulation, the force field should be selected properly according to different study objects. The method of ReaxFF reactive force field molecular dynamics simulation (ReaxFF MD) developed by van Duin etc. can simulate bond formation and breaking between atoms in a chemical process [14–16]. This could lead to a novel and effective way to study the bombardment between AO and polymers.

ReaxFF is trained against quantum mechanics principles but maintains computational efficiency similar to classical force fields, enabling simulations of large-scale systems over extended time frames [15,16]. It has been successfully applied to the research of material pyrolysis [17,18], catalysis [19,20], atomic layer deposition [15,21], carbonization processes [22–26], etc., with ideal achievement. There are also many reports, just like this paper, that have used ReaxFF MD to investigate the impact of AO on polymeric materials in spacecraft, and similarly, they have obtained reliable results. For example, van Duin et al. [27] studied the polyimide materials Kapton, Kapton-POSS, Teflon and amorphous silica for their mass loss, temperature variation, degradation products and erosion yield using the method of ReaxFF for the first time. In their study, they obtained a pleasant consistency between the simulation and the test results; under the same circumstances of AO impact, Kapton had the largest mass loss, while amorphous silica stayed mostly stable. The erosion yield of Kapton and Teflon in the simulation were generally in accordance with the results of the Materials International Space Station Experiment 2 (MISSE 2). Zeng et al. [28] found that the resistance of PVDF to AO is not that outstanding after their study of the AO reaction with PVDF, FP-POSS and their compounds using the method of ReaxFF. However, the capability to withstand AO attack is remarkably enhanced after adding some FP-POSS into PVDF, which is related to the Si-O bond in the FP-POSS. Rahmani et al. [29] compounded POSS, graphene and carbon nanotubes into Kapton, respectively, in which they observed prominent AO resistance improvement after the random compounding of graphene and carbon nanotubes with Kapton. It is worth mentioning that, just recently, Fu et al. [30] studied the polyethylene terephthalate (PET)-AO interaction based on the background engineering requirements in which the PET-AO erosion yield released is rational compared to that from MISSE 2. Although they all demonstrate the promising prospect and potential of ReaxFF MD usage in the analyzation of AO effects on space materials, there are still some defects that need further discussion; all of the simulations are conducted at the initial material temperature of 300 K with a fixed AO bombardment frequency of 5 AO/ps, whereas, in reality, the temperature varies largely in space for the material on the spacecrafts orbiting in the LEO, as well as the AO flux density, which is polytropic depending (but not only) on the altitude of the orbit. Therefore, it is necessary to have a further discussion on the effects and mechanism of temperature and the AO flux density impact on the AO interaction with materials.

In this work, we selected Upilex-S as our research object. Upilex-S is a heat-resistant polyimide film that is the product of the polycondensation reaction between biphenyl tetracarboxylic dianhydride (BPDA) monomers and a diamine. It is an alternative option with excellent performance among polyimide materials. Compared to the most extensively used polyimide material Kapton, Upilex-S has a distinctive molecular structure, but they are both constructed merely by the atoms of hydrogen, nitrogen, carbon and oxygen. The number of atoms in the monomer structure of both materials is the same, except for one oxygen atom less in Upilex-S than in Kapton. Yet, Kapton is more popular around the whole astronautical industry, although Upilex-S possesses a comparatively lower erosion yield, which is revealed from the data of MISSE 2 [31]; this raises our great interest to further explore the process of AO bombardment of the Upilex-S material using ReaxFF MD. Above all, we first established an AO interaction model, and then studied the mechanism of temperature and AO flux density effects on the interaction via bombardment-induced temperature change, mass loss, interaction product, erosion yield, AO number density

deposed onto the matrix, etc., so as to provide technical references for the astronautical applications of the Upilex-S material.

## 2. Computational Details

When we look through the whole process of AO interaction with polymers, bond formation and breaking between atoms should be taken into consideration, where the traditional polymer force fields like COMPASS, CVFF, CHARMM, and AMBER are not appropriate for this study. Here, we adopt the ReaxFF force field proposed by van Duin to conduct the study. ReaxFF verifies the connectivity between atoms using the distance-decided bond order, which is updated at every running iteration in MD, thus providing the allowance of bond formation and breaking. Similar to the empirical non-reactive force field, ReaxFF divides the total system energy into the two parts of bond interaction and nonbond interaction. All the connectivity-dependent interactions are related to bond order, whereas the nonbond interactions such as coulomb and van der Waals are calculated in every atom couple without taking connectivity into consideration. The shielding item is utilized for eliminating any excessive nonbond interactions. A detailed introduction to the ReaxFF force field can be found in the References [14–16]. In this work, ReaxFF parameters raised by Rahnamoun and van Duin, which embrace the H, N, C and O elements, were used in the simulation of the AO interaction with Upilex-S; these have found a lot of successful application in the AO interaction with polyimide material molecular dynamics simulations [27,28,32], with reliable achievement in research.

In this work, the module of Upilex-S is constructed using 240 Upilex-S monomers. In the following, the establishment of the Upilex-S mode and simulation of AO bombardment are under the conditions of original temperature of 300 K, AO bombardment energy of 4.5 eV and AO bombardment frequency of 5 AO/ps; this is similar to the other contents of the study. Figure 1 illustrates the establishment of the model of Upilex-S. At the first stage, the monomer structure ($C_{22}H_{12}O_4N_2$) of Upilex-S is formed, then the total 240 monomers (9600 atoms in all) are set into a cubic box so as to form an amorphous model with a primary density of 0.1 g/cm$^3$. After that, 2 ns NPT (constant atomic number, N; constant pressure, P; constant temperature, T) simulation under the circumstance of atmospheric pressure at 300 K is conducted, in which the control of pressure and temperature use the Berendsen barostat method and Nosé-Hoover thermostat method [33], respectively. The volume of the cubic box is gradually condensed by the NPT simulation so as to set the density model of Upilex-S stable at 1.333 g/cm$^3$ (close to the actual material density), as shown in Figure 2. The 48.0 Å × 48.0 Å × 56.9 Å Upilex-S model is accomplished in the end.

After the establishment of the initial Upilex-S model, the bombardment simulation is conducted using the lammps software. Firstly, for the subsequent AO insertion, the dimension in the Z direction is extended to enlarge the surface of Upilex-S to expand the vacuum region; the x and y directions of the simulation box are set as periodical boundary conditions, while the z direction is set as a fixed boundary condition; after that, the time step is set to be 0.1 fs. Then, the energy of the Upilex-S model is minimized using the conjugate gradient method [34] to conduct a 20 ps NVT (constant atomic number, N; constant volume, V; constant temperature, T) simulation at 300 K, until reaching a dynamic balanced structure at around 300 K. Lastly, the system simulation is changed into NVE (constant atomic number, N; constant volume, V; constant energy, E) mode after the dynamic balanced structure is acquired, AO atoms are inserted randomly in each 200 fs (5 AO/ps) interval at a distance of 64 Å from the surface of Upilex-S, and the random vertical insertion of AO is set at a speed of 0.074 Å/fs (~4.5 eV) to conduct the bombardment simulation. To prevent the Upilex-S from "drifting" downward which would affect the accuracy of the simulation, the 10 Å height at the bottom of the Upilex-S model is fixed in the procedure so as to be excluded from the AO interaction. The output of the degradation products is released by the command "reaxff/species" in lammps. The visible atom analysis is demonstrated using the Ovito software. The product statistics separated from the surface are conducted using Python programming.

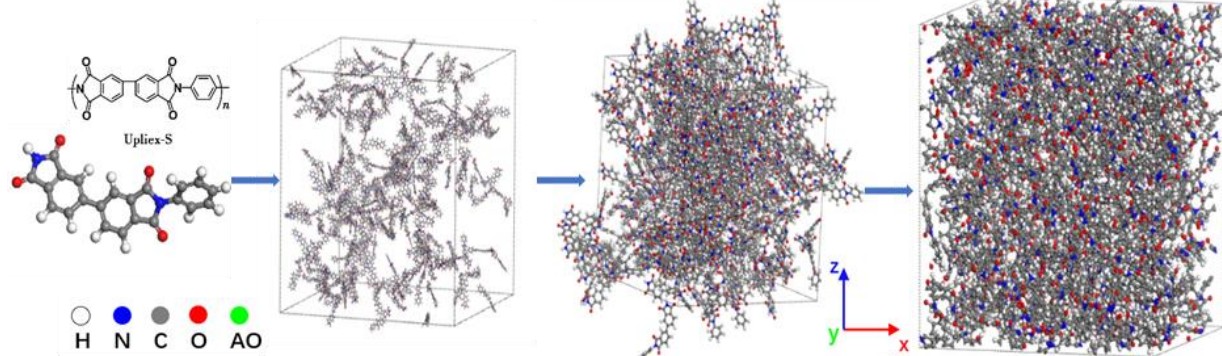

**Figure 1.** Modeling of Upilex-S.

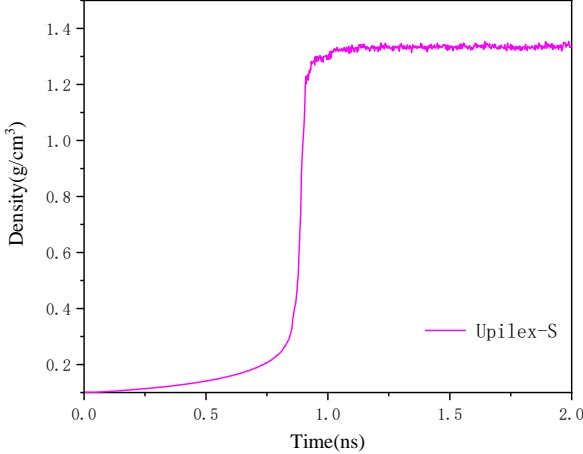

**Figure 2.** Density evolution of Upilex-S model.

## 3. Results and Discussion

### 3.1. The Temperature Effects on the AO Interaction with Upilex-S

When spacecrafts orbit in LEO, their temperature varies between −170–123 °C (103–396 K) [35]. However, according to our research, seldom have we found pervasive studies of temperature effects on the AO interaction with polymers using the ReaxFF MD method. Therefore, for a more comprehensive understanding of the temperature effects, in this section, we examine the mechanism of temperature effects qualitatively by setting the Upilex-S initial temperature at 103, 200, 300, 396, 500 and 600 K (where 103 and 396 K are the extremes in LEO). In the process of AO attack, to avoid the deprived reaction product from the surface which would deviate the subsequent AO bombardment, in this simulation, the product on the Upilex-S surface is removed every 2000 steps (~0.2 ps). The total simulation step lengths of each temperature are all set to be 600,000 steps (~60 ps). The detailed parameter set of the simulations is presented in Table 1.

**Table 1.** Simulation parameters of temperature effects.

| Job \ Parameter | Upilex-S Model | AO Energy (eV) | Initial Temperature (K) | Dose Rate (Atoms/ps) | AO (Atoms) | Time (ps) |
|---|---|---|---|---|---|---|
| 1 | | | 103 | | | |
| 2 | | | 200 | | | |
| 3 | $(C_{22}H_{12}O_4N_2)_{240}$ | 4.5 | 300 | 5 | 300 | 60 |
| 4 | | | 396 | | | |
| 5 | | | 500 | | | |
| 6 | | | 600 | | | |

### 3.1.1. Temperature Variation

High-speed AO bombardment transmits kinetic energy into the material with the result of causing the temperature to increase. Figure 3 demonstrates the temperature variations of Upilex-S under AO bombardment at initial temperatures of 103, 200, 300, 396, 500 and 600 K. It reveals that, along with the AO impact occurring, the temperature increases linearly, and it is comparatively higher at any time than the higher original temperature. A temperature-normalized graph (Figure 4) and a temperature-normalized linear fitting graph (Figure 5) are presented so as to compare the increase rate, which show that it climbs more quickly from the original temperature (103–300 K) while it slows down obviously from a higher initial temperature (396–600 K), without a noteworthy difference between 396, 500 and 600 K. A further analysis of the temperature increase rate result from the linear fitting in Figure 5 was conducted in order to express the relationship between the increase rate and the initial temperature (under the same AO impact condition); this is illustrated as the curve in Figure 6. It shows an apparently inverse relation that the higher the original temperature is, the slower it climbs. This might be induced by the different mechanisms of AO interaction with Upilex-S at different initial temperatures.

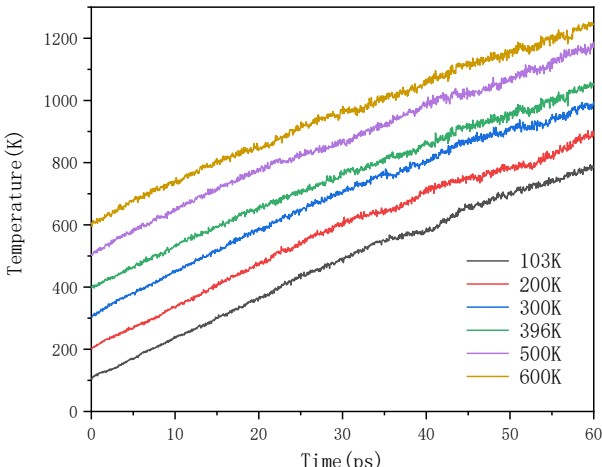

**Figure 3.** Temperature variation of Upilex-S under atomic oxygen impact at different initial temperatures.

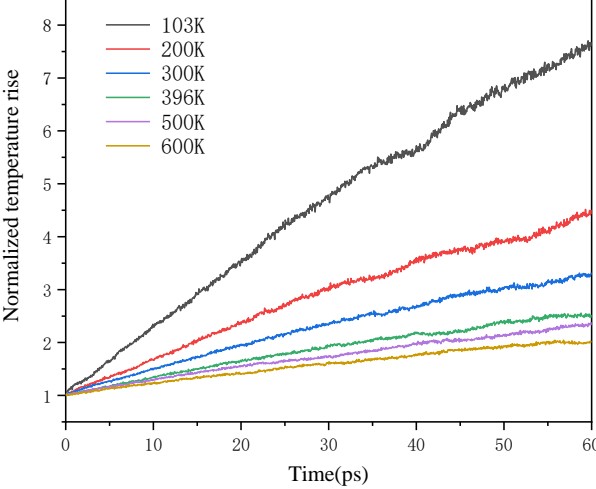

**Figure 4.** Normalized temperature uptrend of Upilex-S under atomic oxygen impact at different initial temperatures. Here, the normalized temperature is the temperature at any given moment divided by the initial temperature.

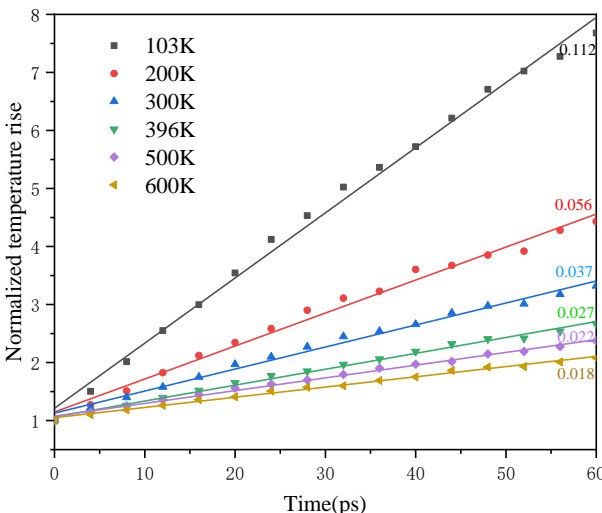

**Figure 5.** Linearly normalized fitting of temperature increase of Upilex-S under atomic oxygen impact at different initial temperatures.

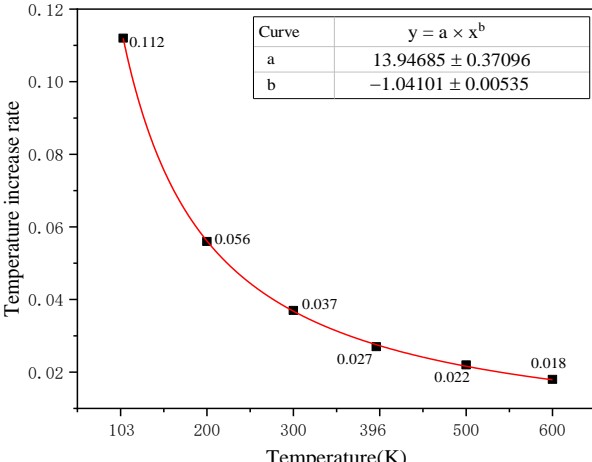

**Figure 6.** Upilex-S initial temperature versus its increase rate.

3.1.2. Mass Loss

In the process of AO impact, the mass loss of Upilex-S was calculated using the residual mass. As illustrated in Figure 7, Upilex-S mass variations at different initial temperatures are curved, which indicate that the mass of Upilex-S is increasing at the beginning of the AO attack, with a generally overlapping tendency. The mass increase is mainly attributed to the AO adsorption onto Upilex-S. This phenomenon is consistent with that from the anterior tests and simulations conducted by other researchers [27,36]. Then, as the AO impact continues, mass loss difference begins to appear between each curve representing different initial temperatures, where the higher it is, the more severe the mass loss is. Among these, the lower original temperature has the minimum mass loss, specifically at 103 K. Figure 8 presents the eventual total mass loss of Upilex-S starting from different temperatures and shows that the mass loss grows as the original temperature rises. In detail, it is not a positive proportional relationship; instead, the distribution of the normalized ratio of total mass loss to initial temperature is basically under the positive proportional line, as shown in Figure 9.

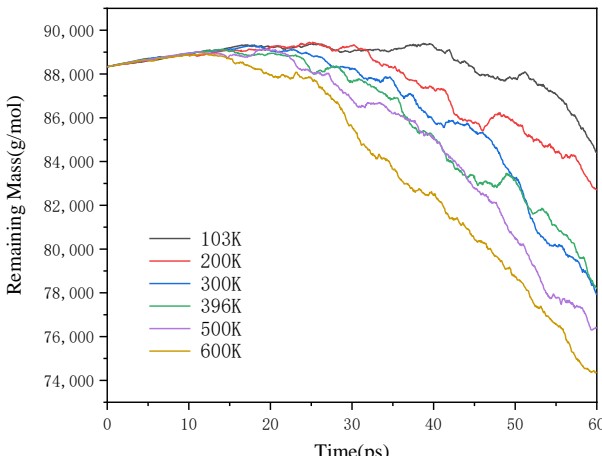

**Figure 7.** Mass changes of Upilex-S during atomic oxygen impact at different initial temperatures.

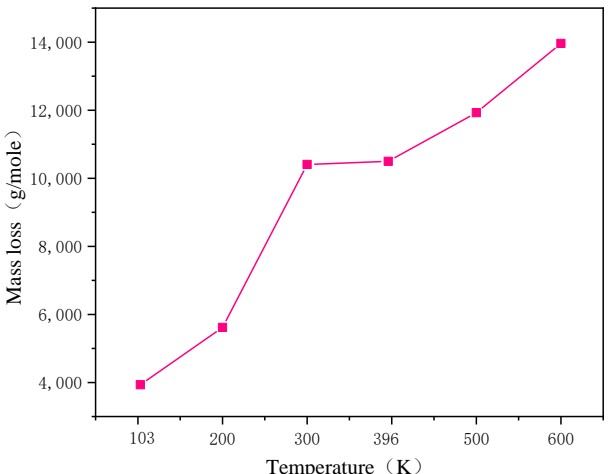

**Figure 8.** Mass loss after AO bombardment of Upilex-S at different initial temperatures (60 ps).

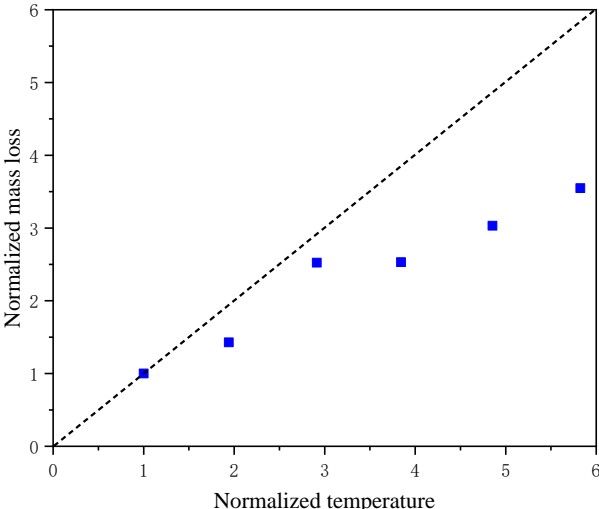

**Figure 9.** Normalized mass loss after AO bombardment of Upilex-S at different initial temperatures.

Temperature is the macroscopic statistical result of massive microscopic atoms' thermal motions. The kinetic energy of the polymer's atoms is relatively lower under cryogenic conditions (especially at 103 K), which results in a stiffer structure of bonds, angles and

dihedrals than at higher temperatures; therefore, more kinetic energy is needed during AO impact to trigger Upilex-S's degradation. That explains the phenomenon of less mass loss induced under conditions of lower original temperature and the same AO attack circumstances. On the other hand, because more active interactive atoms exist at a higher temperature [28], the chemical interaction with AO is easier, which results in a greater mass loss from a higher primary temperature. Nevertheless, further observation of Figure 7 reveals that, in the later period, along with the temperature of Upilex-S increasing, the mass loss of material from a lower initial temperature is still not that severe, which seems to go against the explanation that the higher the temperature is, the more active atoms exist—with more mass loss induced. To dig for the reason underneath the phenomenon for a deeper understanding, a visible AO scattergram and line chart of AO number density distribution in the Upilex-S matrix of 20 ps AO attack at different original temperatures are plotted as illustrated in Figures 10 and 11, respectively. From the perceptual observation of Figure 10, it was found that, at the 20 ps interaction, more and denser AOs deposed onto the matrix in the circumstance of lower original temperature, while the AO number in the matrix decreased obviously and became sparser with the original temperature increase, and it spread more extensively as well. We could also conclude from the AO distribution diagram that the lower the primary temperature is, the higher the peak value of AO distribution in the Upilex-S matrix is, with a narrower width of AO distribution. As the primary temperature increases, the peak value of the AO number density drops, with a more extensive AO distribution; meanwhile, the AO penetration depth is maximized (at 20 ps bombardment, the deepest AO distributions into the Upilex-S matrix under different primary temperatures are 18.86, 19.32, 19.54, 20.21, 21.50 and 22.72 Å, respectively). The information given in Figure 11 is consistent with the presentation in Figure 10. The analysis of Figures 10 and 11 sufficiently indicates that, in the later stage, although the active interactive atoms increase with increasing temperature, the mass loss of Upilex-S tested from a lower temperature is still not that evident. This is because, at the primary stage, more and denser AO deposes onto the Upilex-S attacked at a lower temperature, and although the temperature increases along with the AO sustainable bombarding, more active interaction between atoms in the deeper Upilex-S matrix are still blocked in the subsequent process due to the existence of denser deposition on the surface at the beginning. As a result, the mass loss of Upilex-S tested at a lower temperature is still not that drastic though the temperature increases in the later status. Figure 12 shows the variation in the number of rebounding AO from the surface of Upilex-S during the 0–20 ps impact time at different primary temperatures. It can be seen that, as the primary temperature of Upilex-S increases, the quantity of rebounding AO decreases. This phenomenon may indicate that higher temperatures of Upilex-S allow it to absorb the kinetic energy of AO more effectively.

### 3.1.3. Reaction Products and Erosion Yield

AOs bombard the polymers and interact with the material, so as to isolate the product from the surface which causes the mass loss. To study the type and quantity variation in the reaction product in the test is challenging, but it is necessary for ReaxFF MD. As shown in Figures 13 and 14, we first provide the structural evolution diagrams of Upilex-S at initial temperatures of 103 and 600 K, respectively, during the AO impact process without removing surface reaction products. By comparing Figures 13 and 14, it is found that, under the same AO impact conditions, Upilex-S at a higher initial temperature is more susceptible to erosion by AO. That is to say, this aligns with the observations in Figure 7, where a higher initial temperature results in more severe mass loss. The variation in reaction product and quantity of Upilex-S separated from the surface after AO bombardment beginning at different temperatures is shown in Figure 15. It can be seen that, as the original temperature increases, the quantity of the reaction product increases. The dominant tiny molecule products are HO, $H_2O$, CO, $O_2$, NO and $H_2$, in which the numbers of HO, $O_2$, NO and $H_2$ do not change evidently, whereas the quantities of $H_2O$, CO and other tiny

molecule products classified by carbon atom proportion are larger, along with the primary temperature increases.

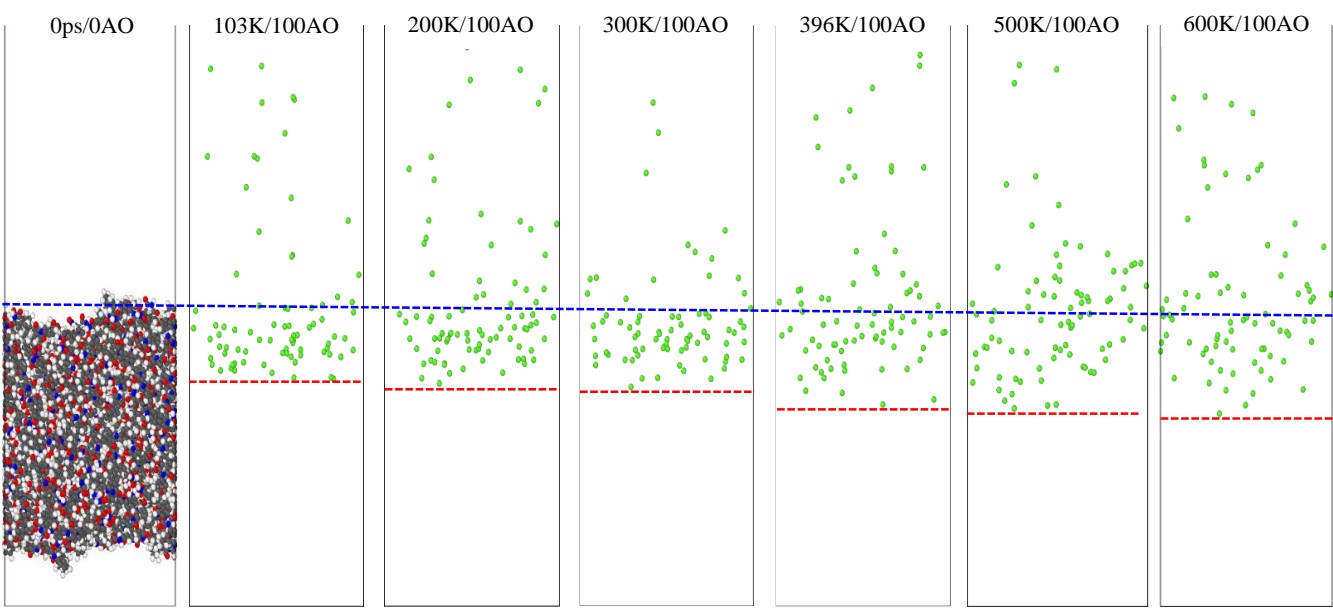

**Figure 10.** AO deposition on the surface of Upilex-S at 20 ps at different initial temperatures.

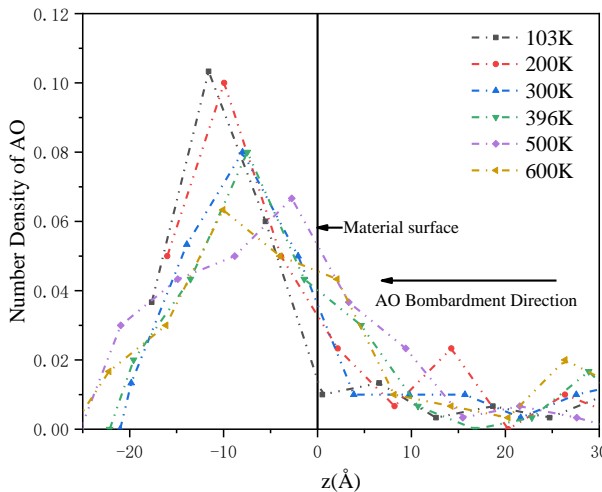

**Figure 11.** AO number density deposed into the Upilex-S matrix at 20 ps at different initial temperatures (z = 0 Å indicates the Upilex-S's surface).

The AO erosion yield refers to the mass or volume loss induced by a single AO after attack. Calculating the volume loss is difficult in a ReaxFF MD study, hence, mass loss calculation at different original temperatures is conducted here. Figure 16 shows the erosion yield variation in the AO interaction with Upilex-S at the same moment beginning at different original temperatures, as shown in the graph; for the same initial temperature, the erosion yield increases as the interaction process continues. The lower the original temperature is, the smaller the erosion yield is. The underlying mechanism is the same as for the analyzation of mass loss; more and denser AO deposed on the Upilex-S surface at a lower temperature at the beginning impedes later AO interactions.

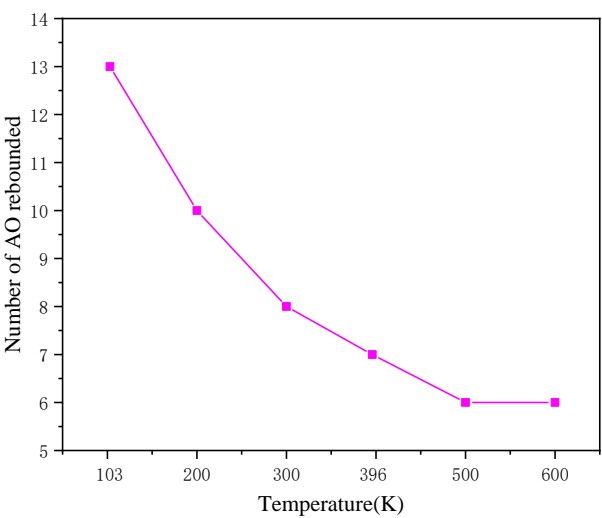

**Figure 12.** Rebounding AO number during the 0–20 ps impact.

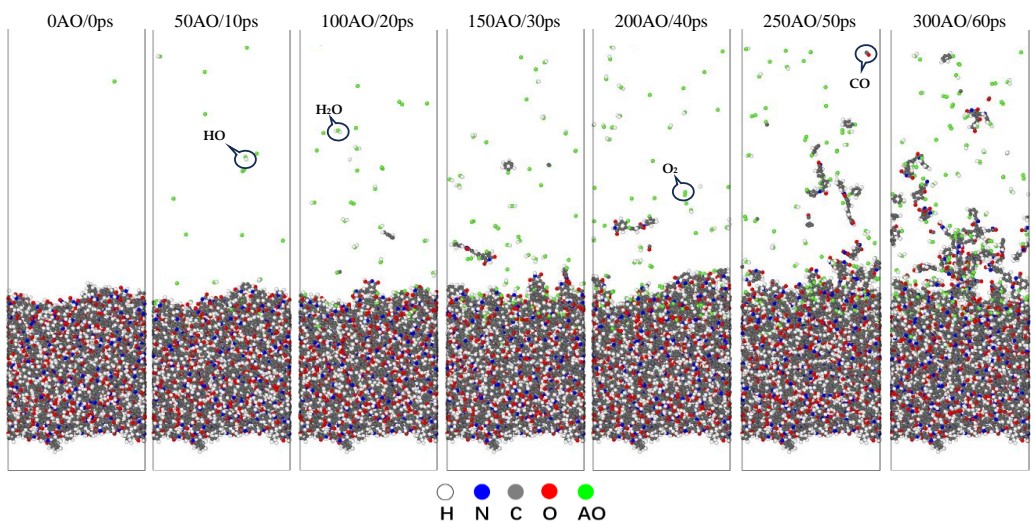

**Figure 13.** Evolution of Upilex-S structure at an initial temperature of 103 K during AO impact.

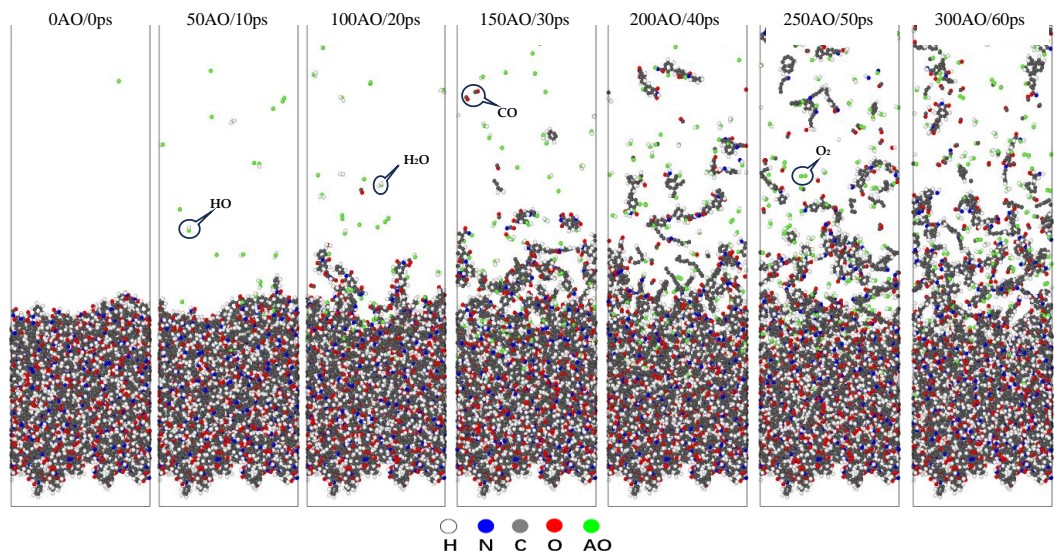

**Figure 14.** Evolution of Upilex-S structure at an initial temperature of 600 K during AO impact.

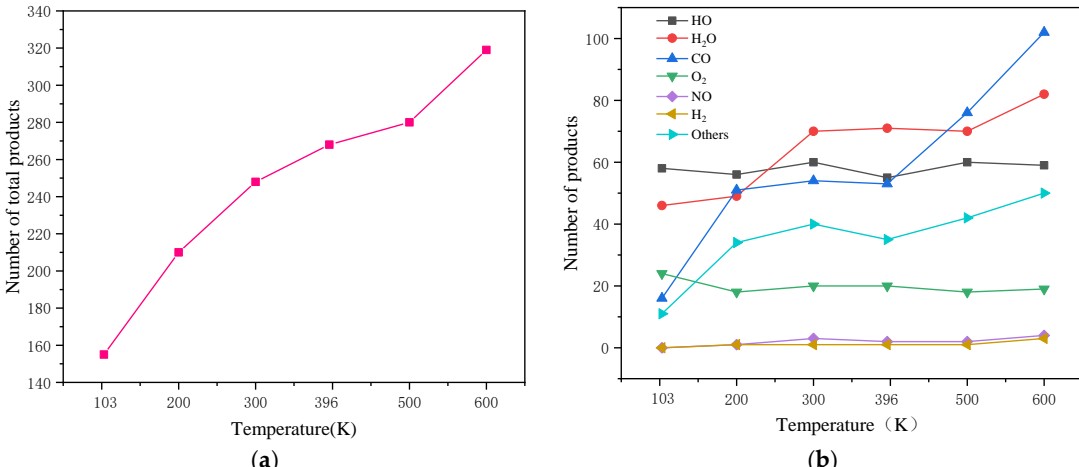

**Figure 15.** Reaction product information of Upilex-S after AO impact at different initial temperatures: (**a**) The total quantity of reaction products changes with the initial temperature. (**b**) The quantity of major small molecule products changes with the initial temperature.

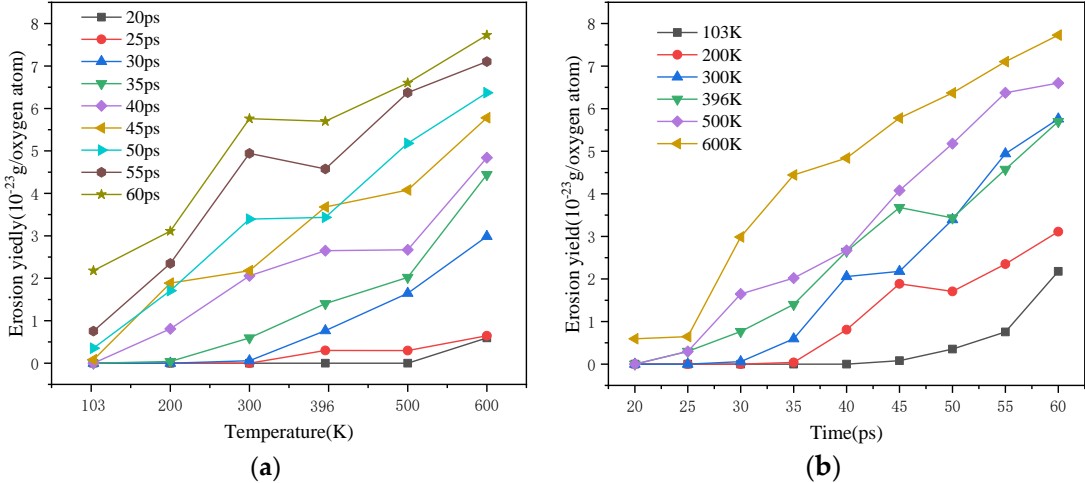

**Figure 16.** Variation in the AO erosion yield of Upilex-S at different moments at different initial temperatures (**a**,**b**).

### 3.2. Effects of Flux Density on the AO Interaction with Upilex-S

The essence of AO flux density variation is the change in AO bombardment frequency in each unit of time. Considering the heavy calculation cost using ReaxFF, we studied the AO flux density effects on Upilex-S qualitatively by changing the AO dose rate (AO bombardment frequency) in this ReaxFF MD simulation, so as to save calculation resources. To conduct this study in the same way as the temperature effects study and to avoid blocking the subsequent AO bombardment, the interaction products released from the surface of Upilex-S are removed every 2000 steps (~0.2 ps). A 30 ps AO bombardment is conducted at every dose rate, and the detailed simulation parameters of flux density are shown in Table 2.

**Table 2.** Dose-rate effects simulation parameters.

| Job | Parameter | Upilex-S Model | Upilex-S Temperature (K) | AO Energy (eV) | Dose Rate (Atoms/ps) | AO (Atoms) | Time (ps) |
|---|---|---|---|---|---|---|---|
| 1 | | | | | 5 | 150 | 30 |
| 2 | | | | | 10 | 300 | 30 |
| 3 | | | | | 15 | 450 | 30 |
| 4 | | $(C_{22}H_{12}O_4N_2)_{240}$ | 300 | 4.5 | 20 | 600 | 30 |
| 5 | | | | | 25 | 750 | 30 |
| 6 | | | | | 30 | 900 | 30 |
| 7 | | | | | 35 | 1050 | 30 |

### 3.2.1. Temperature Variation

The Upilex-S temperature variation under different AO dose rates is shown in Figure 17; it can be seen that the temperature increases linearly as the AO bombardment continues under the same AO dose rate, while under different dose rates, the temperature increases faster under a higher AO dose rate. Figure 18 shows the linearly fitted temperature variation lines representing different dose rates. The slopes of every curve in Figure 18 indicate that the rising rate of temperature is not strictly taking the positive proportion of the dose rate. The normalized temperature–dose rate diagram is drawn in Figure 19 for further demonstration. It indicates that, although the increase rate is consistently beneath the positive proportional line, it is nearly a linear relationship between the temperature-increasing velocity and the dose rate in the lower (5–15 AO/ps) zone. At a higher dose rate, the temperature-increasing velocity slows down, indicating that although more AO bombarded the material at each unit time, as well as more kinetic energy, so as to induce a faster temperature increase, the Upilex-S temperature increase could not be deemed as the effect of the simple summation of single AO bombardments. Further AO dose-rate effects are explored in the following via a Upilex-S mass loss discussion.

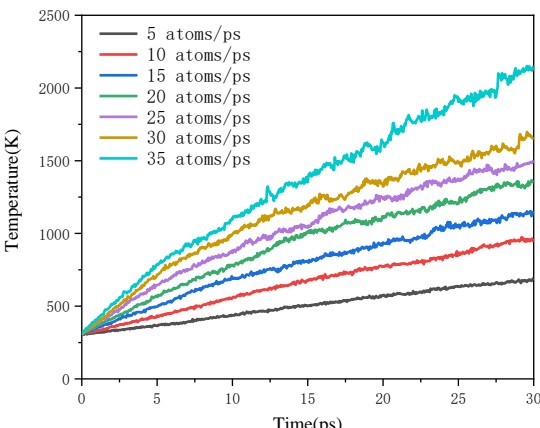

**Figure 17.** The temperature variation in the AO interaction with Upilex-S under different dose rates (the initial temperature is 300 K).

### 3.2.2. Mass Loss

As in the study of the temperature effects, the mass loss variation of Upilex-S under different AO dose rates is calculated using the residue mass. The Upilex-S mass variation under different dose rates is shown in Figure 20. It can be seen that, at the primary stage, the mass of Upilex-S reveals an increasing tendency, which indicates that AO is mainly absorbed onto the Upilex-S surface at the beginning, yet seldom interacts with the material. This is still consistent with the results of the AO interaction with Kapton from other researchers using tests and simulations [27,36]. At the same time, it was found that material under different dose rates resulted in different mass increases; in the initial stage, the higher the dose rate is, the larger the increase in mass is. That might be because of

the mechanism change in the AO erosion of Upilex-S. As the AO bombardment continues, the mass of the material under different dose rates is generally decreasing linearly, which indicates more interaction product in the later stage.

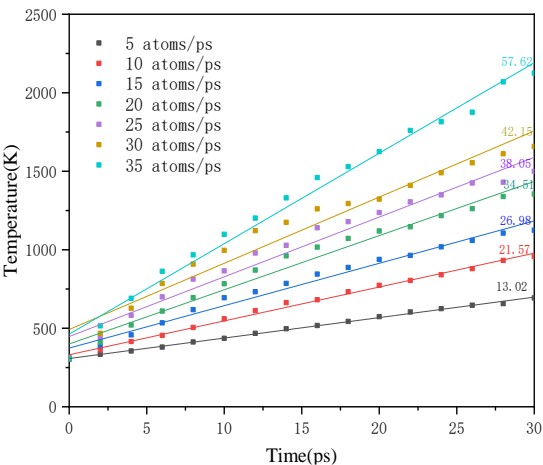

**Figure 18.** Linear fitting of temperature variation in AO interaction with Upilex-S under different dose rates.

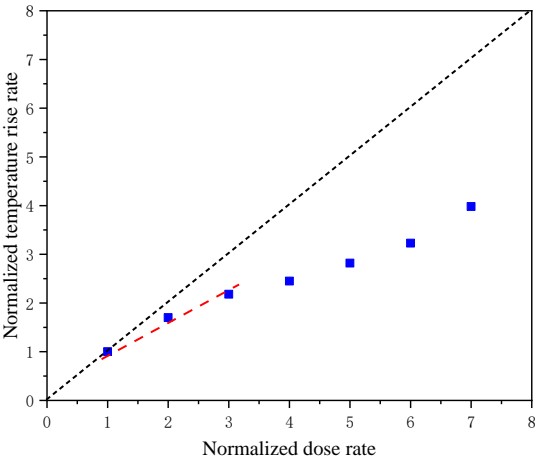

**Figure 19.** Normalized temperature-increasing velocity in the AO bombardment under different dose rates.

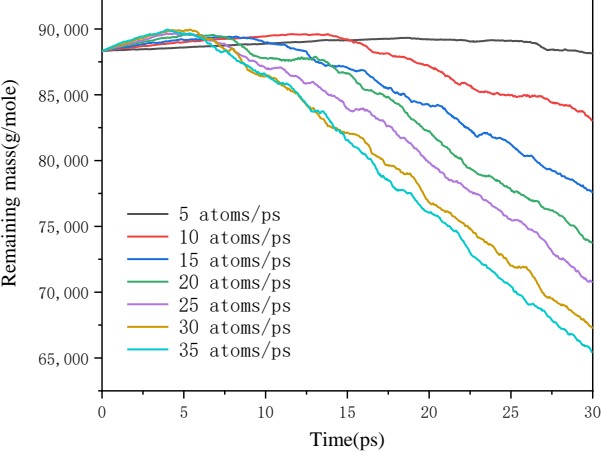

**Figure 20.** Mass changes of Upilex-S during atomic oxygen impact at different dose rates.

Figure 21 shows the total mass loss of Upilex-S under different AO dose rates and indicates that, along with the increase in AO dose rate, the mass loss rate drops. For further analysis on the relationship between mass loss and AO dose rate, a normalized total mass loss-dose rate diagram is drawn in Figure 22. It illustrates that, at the lower dose rate (5–15 AO/ps), although the mass loss increases along with the dose rate increasing linearly, it is not a positive proportional relationship; however, there is a noteworthy enhancement of mass loss under the 10 AO/ps and 15 AO/ps dose rates compared to the 5 AO/ps dose rate that is correlated with the linearly increasing temperature at low dose rates. With more atoms activated due to the temperature increasing, it is easier for the active atoms to interact with AO, which induces a sharp decrease in mass loss [28]. At a higher dose rate (20–35 AO/ps), along with its increase, the mass loss of Upilex-S is mitigated, which seems a little contradictory to physical and chemical intuition; hence, it is necessary to investigate the reason for the mass loss mitigation at a higher dose rate.

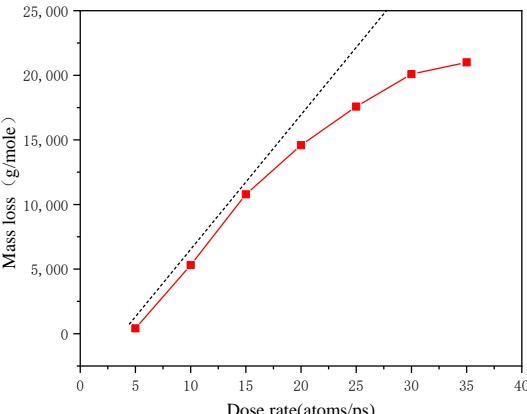

**Figure 21.** Upilex-S total mass loss after different dose rates of AO bombardment.

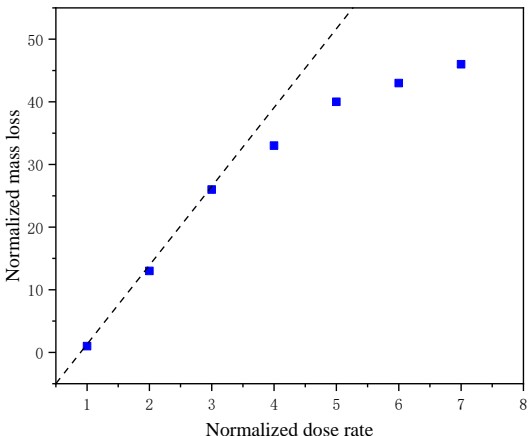

**Figure 22.** Normalized Upilex-S mass loss after different dose rates of AO bombardment.

To investigate the Upilex-S mass loss mitigation at high dose rates, we first counted the percentage of AO interacted with Upilex-S under different dose rates, as shown in Figure 23. Along with the dose rate increasing, the quantity percentage of AO atoms interacting with Upilex-S decreases evidently, explaining the reason why AO-induced mass loss did not increase linearly. But what exactly induced the reduction of AO interacting with Upilex-S at high dose rates? As mentioned above, to prevent the influence of the product on the surface on the subsequent interaction, in this section of simulation it is real-time deleted, so the reduction has nothing to do with it. Therefore, to explore the inner mechanism, we created a visible diagram representing the 5 ps deposition under different dose rates using Ovito, as shown in Figure 24, as we did for the temperature effect study

as well as the AO distribution in the Upilex-S matrix under 5 ps different dose rates. It can be seen from Figure 24 that, at the same moment, the AO quantity and density are all increasing along with the dose rate increasing. It can be seen from Figure 25 that, at the same moment, the peak value of the AO distribution is increasing as the dose rate increases, while the diffusion is weakened; it shows the consistency between Figures 24 and 25. This phenomenon indicates that, since the AO deposition on the surface is fewer and scattered at lower dose rates, the subsequent AO could contact Upilex-S sufficiently, resulting in a large amount of AOs joining the interaction, whereas AO deposition on the surface is more and denser at higher dose rates, which obstructed the subsequent AO contact with the material rigorously and boosted the chance of rebounding after collision at the same time, ending in a decrease in the amount of AO joining the interaction, so as to reduce the mass loss.

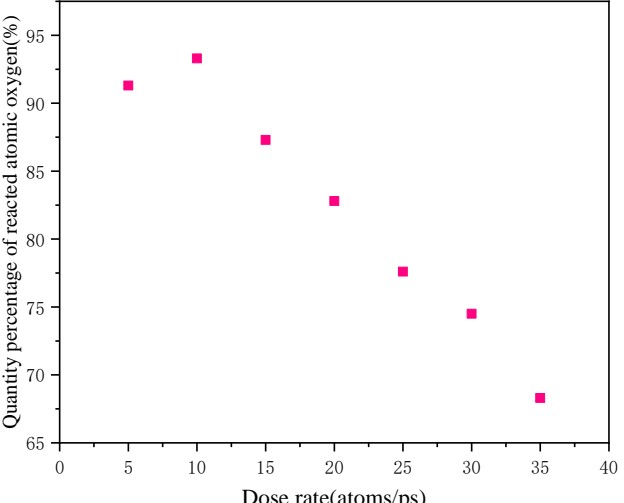

**Figure 23.** The percentage of AO joining the interaction.

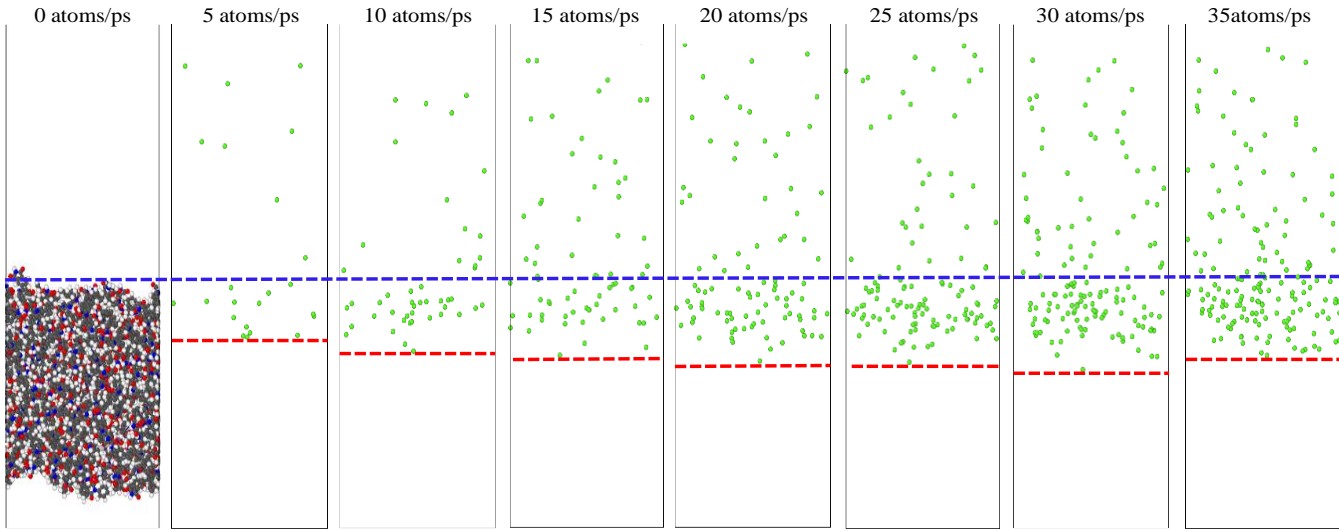

**Figure 24.** The 5 ps AO deposition on the Upilex-S surface under different dose rates.

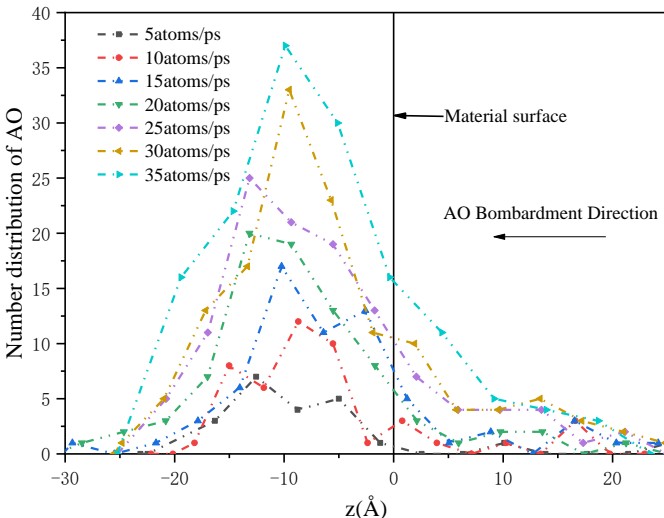

**Figure 25.** The 5 ps AO number density deposed into the Upilex-S matrix under different initial temperatures (z = 0 Å indicates the Upilex-S's surface).

### 3.2.3. Reaction Products and Erosion Yield

Focusing on the dose-rate effect, discussion on the type and amount of variation in interaction products using ReaxFF MD is an essential part as well. As shown in Figure 26, variation in the total erosion increases along with the dose rate, and the primary tiny molecule yields under different dose rates are basically the same as those of HO, $H_2O$, CO, $O_2$, NO and $H_2$, among which the percentage of CO and other internal tiny molecules divided by the C atom increases evidently, whereas the percentage of HO, $H_2O$ and $O_2$ decreases, and $H_2$ and NO show the least variation.

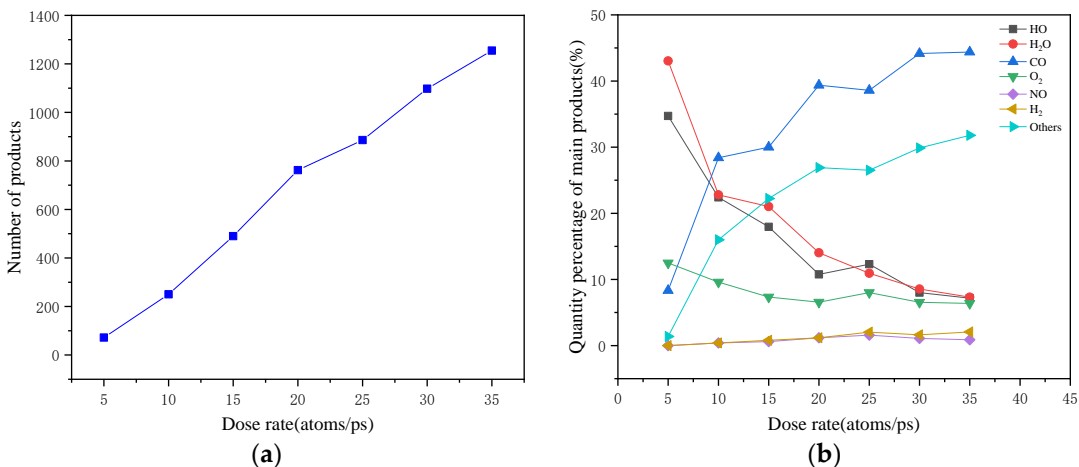

**Figure 26.** Information of erosion products isolated from Upilex-S surface after AO bombardment under different dose rates: (**a**) The total quantity of reaction products changes with the dose rate. (**b**) The proportion of major small molecule products changes with the dose rate.

The erosion yield variation is also obtained by calculating the mass loss due to a single AO. As illustrated in Figure 27 depicting the Upilex-S erosion yield variation at different moments under various dose rates, the erosion yield increases as the AO bombardment continues under the same dose rate. The erosion yields at different moments are almost increasing under lower dose rates (5–15 AO/ps), but the speed of the erosion rate increase is slowing down under higher dose rates (20–35 AO/ps), especially at 20 ps, 25 ps and 30 ps. As mentioned above, the underlying reason is the same; AO deposition on the Upilex-S

surface is more and denser, thus hindering the subsequent interaction and aggrandizing the consequent AO rebounding simultaneously, which slows the increase in erosion yield.

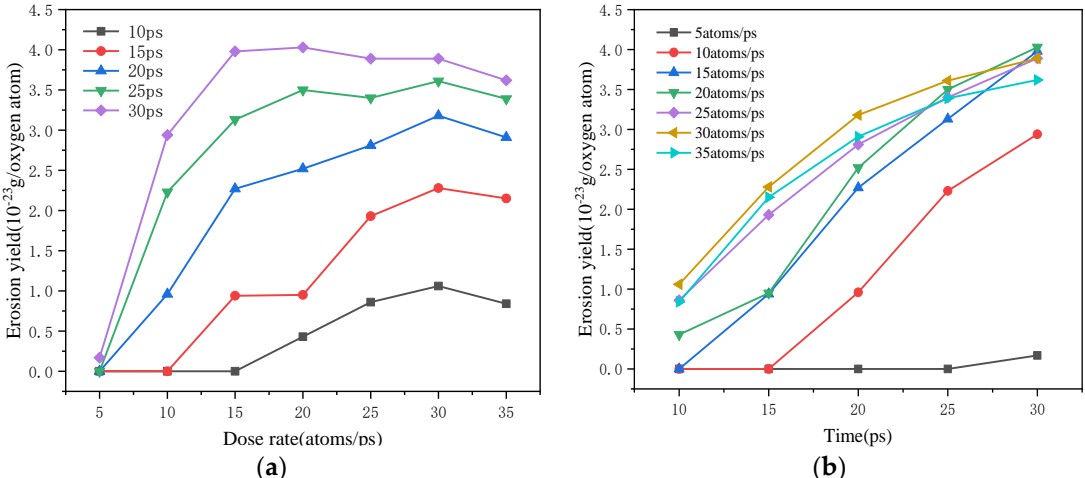

**Figure 27.** Variation at various moments of Upilex-S erosion yield induced by different dose rates (**a**,**b**).

## 4. Conclusions

ReaxFF can simulate the forming and breaking of chemical bonds in a system. In this work, our study is carried out quantitatively via the setting of various original temperatures as well as AO bombardment frequencies to explore the mechanism of temperature effects and dose-rate effects on the AO interaction with Upilex-S. The primary conclusion obtained is as follows:

The effects of AO on Upilex-S are different under different surrounding temperatures. If the heat exchange between the system and external environment is ignored, the Upilex-S temperature increases linearly as the AO impact continues under the same environment. The temperature increase rate is inversely proportional to the Upilex-S surrounding environment. The higher the surrounding temperature is, the slower the temperature of Upilex-S climbs; the damage from AO interaction is insignificant at lower surrounding temperatures because the dense AO deposition on the surface blocks the subsequent erosion; and, as the surrounding temperature goes up, the effects of AO impacts on Upilex-S are exacerbated. Therefore, if Upilex-S is applicated onto an LEO satellite, AO erosion could be alleviated via thermal controlling. This protection method is approximately applicable to other polymer materials vulnerable to AO used in space.

In actual orbit, the AO flux density variation is essentially the same as the dose-rate variation in this work, that is to say, the AO bombardment frequency is different during the same period within the same area. Thus, we could deem it as the damage of different flux density AO is to a different extent. It could be more severe as the flux density increases at the lower stage, while the deposition would increase as the flux rises at the higher stage, which would consequently block further erosion, so as to retard the AO interaction with Upilex-S. Meanwhile, the effects of the AO interaction with Upilex-S under different flux densities could not be comprehended as the simple summation of single AO effects. Therefore, we suggest that different protection measures could be chosen for Upilex-S material applied in different altitude orbiting to enhance the cost savings. Meanwhile, the results indicate that the ground-based test method of shortening the test time-span via increasing the dose rate is approximately not that accurate. This study provides a technical reference and guidance for the improvement of ground-based AO test methods and increasing the accuracy of test results.

**Author Contributions:** Conceptualization, S.Q. and L.J.; methodology, S.Q.; H.J. and L.J; software, S.Q.; validation, S.Q., R.Z. and Y.L.; formal analysis, S.Q.; investigation, S.Q., T.L. and Y.X.; resources, S.Q.; data curation, S.Q. and R.Z.; writing—original draft preparation, S.Q.; writing—review and editing, S.Q. and R.Z.; visualization, S.Q. and Y.L.; supervision, L.J., T.L. and Y.X.; project administration, H.J. and L.J. All authors have read and agreed to the published version of the manuscript.

**Funding:** This research received no external funding.

**Institutional Review Board Statement:** Not applicable.

**Informed Consent Statement:** Not applicable.

**Data Availability Statement:** Not applicable.

**Conflicts of Interest:** The authors declare no conflict of interest.

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
