# Peer review of "Study on the Effects and Mechanism of Temperature and AO Flux Density in the AO Interaction with Upilex-S Using the ReaxFF Method"

_coatings, doi:10.3390/coatings13091586_

Round 1

Reviewer 1 Report

Coatings (ISSN 2079-6412)

coatings-2541731

Study on the effects and mechanism of temperature and AO flux density in the AO interaction with Upilex-S using ReaxFF method by Shiying Qiao , Haifu Jiang , Ruiqiong Zhai , Yuming Liu , Tao Li , Yanlin Xu , Lixiang Jiang. Section:Surface Engineering for Energy Harvesting, Conversion, and StorageSpecial Issue: Micro-Nano Optics and Its ApplicationsIn this article explore the mechanism of temperature effects and dose rate effects on the interaction with Upilex-S and also explore effects of AO to Upilex-S is different under different surrounding temperature. The abstract does not introduce or explain what was done in this article. “Our study could be provided as a technical reference for the wide-usage of Upilex-S on the spacecraft. “ This is certainly not a technical reference. The introduction does not introduce the problem that leads to the analysis. The quantities discussed in the work were not introduced to the level of scientific work. The methodology of analysis and work is not clearly and precisely introduced. The analysis is quite superficial and illogical. The English language is not clear and unambiguous. No references are given to chemical, physical, mathematical research that exists in the field of AO-polymer interactions. Referencing is insufficient. The reasons for the expected changes depending on the parameter are not specified. Unfortunately, although the topic is interesting and important, due to the way of presentation, I propose the paper to be rejected. Especially a publication like this is not for the special Issue where optical characteristics at the micro-nano level are considered

The English language is not clear and unambiguous. 

Author Response

Dear Reviewer,

Thank you for your
precious suggestions and criticisms giving to my manuscript. I sincerely appreciate the time and support that you provide to my work.

Based
on your suggestions and comments, I have carried out necessary revisions to my manuscript. Additionally, I have taken into account the recommendations and queries from Reviewer 2 and Reviewer 3, ensuring that my paper is more refined and accurate. The decision regarding the topic under which this manuscript will be published is determined by the editorial
team of the Coatings journal, and I hope you can understand
.

Once again, I sincerely appreciate the comprehensive and insightful questions and suggestions you provided.

S
incerely,

Reviewer 2 Report

The authors present an excellent work on the effects and mechanism of temperature and AO flux density in the AO interaction with Upilex-S using ReaxFF method

I recommend this for publication with a minor revision as follows:

First of all, the title could not explain the concepts of the manuscript contents for readers, explicitly and must be modified in better format.

The authors should address the following issues prior to publication:

1. Compared with other section of manuscript, introduction is short and some further explanation and presented more details of background are needed.

2. Both the number of references and also the references from recent years are too much low (at least 50% references should be from 5 recent years

3. In many places the abbreviations including IUPAC name must be explain for readers, such as: Upilex-S, Kapton, MD,  ReaxFF (reactive force field) and several other items

4. In table 1, in the column 2, Upilex-S model makes the reader confused due it is not a model is better write Upilex-S material or something else.  It is notable Upilex is just a heat-resistant polyimide film that is the product of the polycondensation reaction between biphenyl tetracarboxylic dianhydride (BPDA) monomers and a diamine. It is better this explanation is also add in the related text.

5. Authors believe, AO interaction with polymers with consideration the traditional polymers force field like COMPASS, CVFF, CHARMM, AMBER are not appropriate for this study. Why? I don’t believe it in fact charmm is strong software for most of MM or MD calculations. The authors should be explain the advantages of ReaxFF compared with other force fields

6. Although Upilex-S material has astronautical applications, but in fact the foundation of manuscript is a MD calculation not only space or LEO, the explanation of authors about MD calculation of the system is very low. I cannot see the optimization data for Upilex-S that is an essential factor for any simulation of using polymer macro molecules, or for MD calculations, atomic partial charge is one of the most important parameters. Why the authors did not considerate this parameter?

7. Since ReaxFF was developed to bridge the gap between quantum (QC) and empirical force field (EFF) based computational chemical methods the minimization data of the system for the monomer interaction with AO in various situation of temperature and pressure based on ab-initio calculation is needed.

8. What does mean the sentence in line 358- which is written as follows?

As it mentioned above, to prevent from the influence of product on the surface to the consequent interaction, in this section of simulation, it has been real-timely deleted, so the reduction has nothing to do with it. (More explanation is needed)

9- Figs 1 and 10 are unclear

14- What do we learn from Figure 6, figure 13 and figure 14? Some more explanation is needed?

I recommend publishing the work after minor revision.

Minor editing of English language required

Author Response

 Dear Reviewer, 

I have already responded to all of your comments and suggestions in the attached file.

Sincerely,

Reviewer 3 Report

The authors present very interesting simulation data for the oxygen bombardment of polymeric surfaces using the novel ReaxFF force field. The latter is capable to diagnose the bond breakage and the formation of new bonds in polymers, and thus to analyze the degradation rate of the polymer and the reaction products of the bombardment. Extensive data is collected on the degradation of the polymer, the T effects and the bombardment dosage on it.

Below are my comments:
1. Figure caption 4. Clearly state that normalized T is T(t)/T_0.
2. Figure caption 8, provide the simulation time at which the data was collected.
3. Page 8, line 238, explain what you mean under the term of diffusion. Is it how deep the AO enters the surface before it is captured/stopped by the polymer?
4. Talking about the diffusion, I think showing MSD curves will greatly improve the understanding of the T effects on the bombardment data.
5. Figure 10, what is the origin of the AO above the polymer surface: is it the bombarding agents, or the scattered back AO from the surface? In both cases I would like to see they being pictured in different colors. What happens to the rebounded AO, are they removed from the cell, or they keep flowing backward  towards positive z values?
6. Figure 12 claims that there is less rebounded AO as T increases, while Figure 10 shows that more AO is rebounded. I just tried to count the number of dots above the surface. Am I wrong?
7. Seemingly, there are more scattered back AO near the surface at higher T, seen from Figure 10. Please give more details why this happens and has this something to do with the fact that higher T surface can absorb more kinetic energy from the bombarding AO.
8. Section 3.1.3. How you identify which atomic groups are the reaction products? For example, if you capture an AO at some depth in the polymer, then you have to apply some sort of criteria according to which you identify the break of bonds in the monomer and the formation of new bond between AO and some atoms. Citing ref.14 on page 3 is not enough, at least a one-paragraph information is needed for completeness of this work.
9. Which parts of the monomers are more vulnerable for erosion? Can you provide a snapshot picture showing initial monomer, and the bombarded monomer considered eroded and the reaction product. Without this picture the common words that AO erodes the polymer is not scientifically sound.
10. Figure 14, one of the lines here should be the same shown in Fig.8, which one?
11. Figure 15, what is the initial T, and put that in figure capture. Please try to give a complete info about the T and time at which the data shown in all Figures.
12. Fig 22, similar correction as in comment 5, different colors for AO captured and rebounded, and show the maximal capture distance from the surface as it was done in Figure 10.
13. Is AO charged? Explain how the charge neutrality condition in simulations was controlled.

Minor typos.
1. Page 2, line 65, it is Rahmani, not Rahmnai.
2. Page 1, line 29, should be 'which largely exists'.
3. Page 3, line 107, Reference 1414 should be 14.
4. Page 4, line 159, 2000 steps is 0.2 ps, not 0.2 fs. The same on page 11, line 292.

In conclusion, the subject of the manuscript is indeed belongs to the interesting and undeveloped research area and I believe its publication will inspire others to dig more to find better materials for coating of crafts in orbital missions. Nevertheless, the description of the observed data is not clean and should be improved, especially the break of chemical bonds and the formation of reaction products next to the initial monomer. What criteria were used should be discussed in details. More information is needed about what kind of damage the monomer suffers. I recommend the authors to amend their work and resubmit it again to the Coating Journal.

It is OK.

Author Response

(The authors gave the same response as above.)

Round 2

Reviewer 3 Report

I am fully satisfied with the response of the authors.  In its current form the modified manuscript is publishable as is in the Coating journal.